# Investigating the Impact of Lunar Rover Structure and Lunar Surface Characteristics on Antenna Performance

**DOI:** 10.3390/s24165361

**Published:** 2024-08-19

**Authors:** Rida Gadhafi, Elham Serria, Sara AlMaeeni, Husameldin Mukhtar, Raed Abd-Alhameed, Wathiq Mansoor

**Affiliations:** 1College of Engineering and IT, University of Dubai, Dubai 14143, United Arab Emirates; eserria@ud.ac.ae (E.S.); hhadam@ud.ac.ae (H.M.); wmansoor@ud.ac.ae (W.M.); 2Space Robotics Lab, Mohammed Bin Rashid Space Centre, Dubai 211833, United Arab Emirates; sara.almaeeni@mbrsc.ae; 3Faculty of Engineering and Informatics, University of Bradford, Bradford BD7 IDP, UK; a.a.a.abd@bradford.ac.uk

**Keywords:** lunar communication, sleeve balanced antenna, lunar antennas, lunar surface propagation, lunar propagation channels, Rashid rover

## Abstract

This article explores the influence of lunar regolith and rover structure, such as mast design and material composition, on antenna parameters. It focuses on the distinctive difficulties of communication in the lunar environment, which need specialized antenna solutions. This study specifically examines the performance of antennas on the lunar Rashid rover within the Atlas crater, a landing site on the moon, considering two antenna types: a sleeve dipole antenna and an all-metal patch antenna. Thermal analyses reveal temperatures in the Atlas crater can exceed 80 °C during lunar mid-day. The findings highlight the effect of different materials used as thermal coatings for Rashid rover antennas, as well as the influence of rover materials on antenna performance. Furthermore, this study extends to analyze the conductivity and depth of lunar regolith within the Atlas crater. Given the critical role of antennas in wireless communication, understanding how lunar regolith properties affect antenna performance is essential. This research contributes to the creation of a strong communication system for the Rashid rover and future lunar missions by considering the features of the lunar regolith in addition to the rover’s size and material attributes.

## 1. Introduction

The moon has historically intrigued humanity’s quest for understanding celestial bodies. Early observers noted dark features on the lunar surface, speculating about a potential terrestrial presence. In 1609, Thomas Hariot and Galileo Galilei’s telescopic observations revealed that these dark regions had smoother terrain compared to the surrounding lunar landscape. Johannes Kepler later classified these areas as “maria” (seas) and “terrae” (lands). Researchers found that the lunar maria, at lower altitudes, are primarily composed of dark basaltic materials, while the higher-altitude lunar highlands (terrae) are mostly made of anorthosite [1,2]. In 1958, following the Soviet launch of Sputnik, President Eisenhower established NASA to address aeronautical and astronautical research. NASA’s mandate includes exploring both atmospheric and space environments [3]. On 21 July 1969, Neil Armstrong made history with the first moonwalk [4]. As a result, lunar missions have been strategically planned in three distinct phases: the initial phase involves deploying robotic missions for preparatory work, followed by the establishment of a lunar outpost with human presence in the second phase, and ultimately, the ambitious goal of constructing a permanently inhabited lunar base in the third phase [5]. NASA’s Apollo program aimed to explore the Moon through a series of missions. Apollo 7 and 9 orbited Earth to test the command and lunar modules. Apollo 8 and 10 orbited the Moon, conducting tests and capturing lunar surface images. Although Apollo 13 did not land on the Moon, it returned crucial lunar photographs. The successful missions Apollo 11, 12, 14, 15, 16, and 17 achieved lunar landings and safe returns to Earth [6]. Following the Apollo program spanning from 1969 to 1972, the missions persistently transmitted data to Earth until the stopping of other onboard instruments [7]. NASA, in collaboration with commercial partners and international space agencies like the European Space Agency (ESA) and Italian Space Agency (ASI), is actively engaged in plans to return humans to the Moon and advance towards eventual Mars exploration. The Artemis mission, launched in 2017 as part of NASA’s Moon-to-Mars program, initiates this effort. These missions aim to coordinate lunar surface activities, establish sustained human habitation on the Moon, and drive technological advancements essential for future Mars exploration. The ultimate goal is to create a sustainable lunar presence as a critical step towards future human missions to Mars and beyond [8,9]. NASA began its lunar exploration efforts with the uncrewed Artemis 1 mission in December 2022. This mission successfully orbited the moon, carrying three anthropomorphic mannequins, various tree seeds, and multiple experiments, payloads, and CubeSats. The upcoming Artemis 2, scheduled for around November 2024, will conduct a lunar flyby with a four-person crew. Following this, the Artemis 3 mission, planned for 2025 or 2026, aims to land at the lunar south pole using the advanced SpaceX Starship as the primary spacecraft [10].

Many nations, encompassing Japan, Russia, the United States, South Korea, India, and the United Arab Emirates, are actively pursuing lunar missions with the shared objective of comprehensively exploring and understanding the Moon’s surface and distinctive features. Sponsored by the Mohammed Bin Rashid Space Centre (MBRSC), the Emirates Lunar Mission (ELM) stands as a dedicated endeavor to plan and execute the United Arab Emirates’ first robotic lunar exploration [11]. At its core, the ELM features the Rashid rover, poised for lunar deployment with a period spanning one lunar day, corresponding to approximately 14.5 Earth days. Equipped with advanced tools and experiments, the Rashid rover is carefully designed to study how the lunar surface changes. This helps us gain important insights needed for future missions to surfaces without atmospheres in our solar system [12]. The initial challenge in any lunar mission is selecting a specific landing site on the lunar surface. The chosen area should possess characteristics that qualify for the analysis and careful examination for the collection of lunar surface data [13]. For the Emirates Lunar Mission, the selected landing site is the Atlas crater. This decision is based on the unique characteristics of the crater, which will be carefully considered in all experiments. The Atlas crater is situated northeast of Mare Serenitatis and southeast of Mare Frigoris, with coordinates at longitude 44.4° E and latitude 46.7° N. It boasts an 88 km diameter and dates back 3.2–3.8 billion years. The crater is circular and enclosed by an intricately terraced rim wall. Significantly, it descends to a depth of 2 km and showcases an intricate floor adorned with hills and cracks [14,15]. The primary function of the Rashid rover is to facilitate data exchange with the ground station, a task accomplished through its two communication subsystems. The primary communication subsystem, accompanied by a sleeve dipole antenna, is dedicated to data exchange with the lander. In contrast, the secondary subsystem, which is a patch antenna, is responsible for facilitating data exchange between the rover and Earth [12]. Given the complexity of these missions and the critical role of antennas in communication systems, it is crucial to understand their behavior in the lunar environment before deployment. Performance characterization of antennas in this setting is highly valuable.

A comprehensive overview of the diverse types of antennas used in lunar applications has previously been reported in [11], where lunar antennas in terms of type and application—whether for research or real missions—were examined. These antenna designs include microstrip monopole antennas [16], fractal antennas [16], Vivaldi antennas [17], bow-tie antennas [18], inf-IRA antennas [19], all-metal reflector antennas [20], polyimide film antennas [21], and both dipole and monopole antennas [18,22]. Among these, only a few are meant for real missions. Antennas designed for specific missions should offer advantages such as a compact profile, easy installation, low power requirements, and durability, all while being protected by a radome. Therefore, understanding the effect of radomes on antenna performance is crucial. Additionally, the impact of lossy ground on antenna performance is often overlooked, except in [23], where the authors analyzed the effect of lunar terrain on signal propagation using the Geometrical Theory of Diffraction (GTD) to account for reflections and diffractions from three-dimensional lunar terrain and objects around the antenna. Also, understanding the impact of lunar rovers on antenna behavior is crucial but rarely addressed in the literature. With the recent boom in space applications, particularly lunar communications, addressing the challenges antennas face in the lunar environment is increasingly significant. It is inadequate to assess stand-alone antenna performances since they are always integrated into structures like rovers. Lunar soils, with varying permittivity and conductivity, directly affect antenna radiation patterns, causing signal oscillations and deep nulls due to strong ground reflections. Therefore, it is essential to consider these constraints and thoroughly understand each component’s behavior to ensure reliable communication, as post-launch adjustments are often difficult in real missions.

An initial study presented an analysis of a sleeve dipole antenna in the presence of lunar regolith and a rover [24]. Another study explored the performance of this antenna with various coating materials [25]. In this current study, we provide a comprehensive analysis of the influence factors on the performance of two specific antennas planned for a lunar rover mission. This includes detailed simulations and the consideration of multiple factors such as lunar regolith, rover structure, and antenna coatings. Here, we utilized a novel combination of EM simulations to examine the performance under realistic mission conditions. This approach provides a thorough understanding of the environmental and material impacts on antenna performance. For this purpose, this article builds on earlier analyses by incorporating detailed characteristics of the landing site, of the Atlas crater, and of the rover’s structure into the simulations. It further assesses the performance of coating materials on a sleeve antenna integrated into the rover structure under the presence of lunar regolith of the Atlas crater. For that, a 3D model of the lunar regolith, rover, and antennas integrated into the rover was modeled using an electromagnetic simulation tool. Additionally, it explains the design of an all-metal patch antenna and evaluates its effectiveness when integrated into the rover structure in the lunar environment. This study also examines the behavior of antennas while rotating to different angles, a mission requirement to satisfy the pointing mechanism and ensure effective communication between the Rashid rover in the Atlas crater and ground stations on Earth. We anticipate that this systematic approach, focusing on more realistic scenarios, will provide valuable insights to the scientific community in developing antennas for lunar communications, addressing constraints that were previously overlooked. We aim to provide a guide for the antenna research community interested in designing or analyzing antennas for lunar applications. Furthermore, the findings from our study have significant practical implications for the design and optimization of communication systems for lunar missions. The insights gained can inform future lunar mission designs and improve communication reliability under harsh lunar conditions.

This study is structured as follows: Section 2 provides a description of the Rashid rover, Section 3 discusses lunar regolith, and Section 4 focuses on antenna designs. Next, we provide the findings in Section 5, followed by a discussion in Section 6 and, subsequently, provide a concise conclusion in Section 7.

## 2. Rashid Rover

The Rashid rover stands as the preliminary project of the Emirates Lunar Mission (ELM), embarking on lunar exploration within a single lunar day. Its primary goal is to showcase the national proficiency in developing technologies for lunar surface exploration. The secondary objective involves the systematic execution of scientific investigations on the lunar surface [26,27]. To facilitate data collection and perform lunar surface operations, the Rashid rover is equipped with diverse subsystems, including structural systems, a mobility mast and gimbal, and subsystems for science, imaging, thermal measurements, power, and avionics. Ensuring temperature control for each of these subsystems is imperative for their effective functioning in carrying out their respective tasks [28]. The Rashid rover, uniquely tailored for its mission, boasts a designed mass of 10 kg and features specialized instruments. Among its array of capabilities are four distinct types of cameras. The Microscopic Imager (CAM-M) serves as the first to capture high-resolution images of the lunar surface. The second camera, known as the Thermal Imager (CAM-T), is dedicated to conducting comprehensive studies on the thermal properties of lunar surface features. Additionally, the Rashid rover is equipped with two identical px CMOS wide-field cameras, CAM-1 and CAM-2, each covering a substantial field of view at 85 degrees. Supplementing its camera systems, the Rashid rover incorporates cylindrical Langmuir probes, collectively referred to as the Langmuir Probe System (LNG) which are positioned at various heights on the rover’s body; these probes meticulously trace electron densities within a range of approximately 15 cm to 65 cm above the lunar surface. Ultimately, the rover achieves mobility through the utilization of four wheels equipped with grousers, as illustrated in Figure 1 [27,29].

## 3. Lunar Regolith

The lunar regolith is a fine-grained reworked surface deposit up to 10 m deep. Below this layer, coarse-grained polymict ejecta and comminuted melt sheets can be found. In the Emirates Lunar Mission, the decision was made to dispatch the Rashid rover to a region on the moon known as the Atlas crater [30]. This crater, serving as the landing spot for the Rashid rover, boasts a regolith depth of 1.6 m. In the absence of water at this lunar mid-latitude site, lunar soils display a DC electrical conductivity of 10^−14^ S/m, whereas lunar rocks demonstrate a conductivity of 10^−9^ S/m at 300 K in the darkness [25]. Exposure to sunlight triggers a notable increase of over one million times in electrical conductivity for both lunar soils and rocks. Given that the proposed antennas operate in the GHz region, the RF signal penetration is significantly lower compared to MHz frequencies, resulting in a reduced skin depth. Hence, it is imperative to conduct thorough simulation analyses to detect any alterations in radiation patterns or impedance matching that could affect their performance. Furthermore, potential communication losses may arise due to the composition and conductivity of the lunar landing site. Lunar regolith was incorporated into the design beneath the upper panel of the rover, considering the dimensions of its wheels. For the Atlas crater, which has an average regolith depth of 1.6 m, we approximated the regolith as a homogeneous material with varying conductivity values to account for both higher and lower extremes and having a permittivity. Given the pivotal role of antennas in communication systems, understanding the impact of these parameters on antenna performance is crucial.

## 4. Antenna Design

As previously mentioned, the Rashid rover employs a dual-antenna system to cater to its communication requirements. The first antenna, a sleeve dipole antenna, is dedicated solely to primary communication between the rover and the lander (lunar surface communication) with a gain of 2.23 dBi. Based on the link budget calculations for the Rashid mission, which consider a maximum distance of 200 m between the transmitter and receiver and a data rate of 250 kbps, the gain of 2.23 dBi is sufficient to ensure reliable communication. In the event of primary communication failure, the secondary communication subsystem (from rover to earth), utilizing a circularly polarized patch antenna, comes into operation. This secondary communication serves a dual purpose: it serves as a backup for primary communication and also fulfills a crucial mission objective by establishing direct communication with Earth during the second lunar day. Additionally, it functions as an experimental payload to assess the capabilities of an innovative and power-efficient lunar-to-Earth communication system. The secondary antenna operates at two distinct frequencies: 2266 MHz for downlink and 2101.2 MHz for uplink communications, with a bandwidth of 100 kHz and a gain of 6 dBi. The primary purpose of the patch antenna in the Rashid mission is to serve as an experimental payload. For the intended application, our link budget calculations indicate that a gain of 6 dBi is adequate to support a few kbps data rate. This data rate is sufficient for the experimental payload’s communication needs, ensuring effective data transmission within the mission constraints. Commercially available antennas suitable for space missions were employed for this purpose. Simulations were conducted using models closely resembling these antennas in terms of structure, dimensions, and performance. To ensure a fair comparison of antenna performance in surface communication, both antennas were designed to operate at the same frequency and polarization. This allowed for an examination of the behavior of a sleeve dipole antenna as well as all-metal patch antennas in the presence of a rover structure and lunar regolith. This study aims to evaluate these antennas for lunar surface communication. The simulation platform used for these endeavors was CST Microwave Studio (2023 version). Systematic simulations were conducted to comprehend the behavior of the antennas in the lunar environment.

Initially, the performance of standalone antennas was simulated. Subsequently, following mission requirements, the primary antenna was coated with various materials, and its performance was evaluated. To understand the impact of lossy regolith, a regolith model was created in CST, and antennas were placed at various heights above the regolith, observing their performance. Furthermore, the behavior of each antenna when integrated into the rover structure was examined. To achieve this, a detailed model of the Rashid rover was constructed in CST, and antennas were integrated separately. Finally, a comprehensive model was employed with antennas integrated into the rover structure positioned atop the regolith, observing the performance of each antenna. This took into consideration the fact that during the mission, both antennas would never be activated simultaneously.

### 4.1. Sleeve Dipole Antenna Design

The antenna geometry, as shown in Figure 2, represents a predefined configuration tailored to meet the requisite communication characteristics. The sleeve dipole antenna, with a total length of 55.7 mm, comprises an inner conductor and sleeve components and is excited through a wave port, as shown in Figure 2a. In order to match the antenna’s height with the height of commercially available antennas, a dielectric tube (Acrylonitrile Butadiene Styrene (ABS), ε_r_ = 3.2, tan δ = 0.025) with 200 microns was used to cover the inner conductor of the antenna to increase the antenna’s length to 159.5 mm, Figure 2b.

#### Sleeve Dipole Antenna with Coated Materials

Antennas and electrical equipment deployed in the field are inevitably exposed to diverse environmental stressors. The lunar surface, in particular, presents challenges such as temperature extremes, regolith presence, and the potential impact of dust clouds, all of which can compromise antenna integrity and performance. In this context, the coating materials are of paramount significance. Coating materials primarily serve as protective layers for antennas, with their impact on antenna performance considered relatively marginal [31,32].

Coating materials can be applied to antennas through various methods. While some coatings involve the application of paint directly onto the antenna surface, this approach necessitates periodic reapplication of the paint and may be prone to issues such as flaking or chipping. Conversely, alternative coatings are composed of diverse formulations of polymers or synthetic materials, wherein long-chain molecules are intricately bound to attain specific and tailored properties; polytetrafluoroethylene (PTFE) stands out as one of the most renowned polymer coatings widely employed in contemporary industrial applications. The influence of coated materials on antenna performance manifests through several facets, including shifts in resonance response and alterations in radiation pattern gain and beamwidth, as well as the potential for degradation of overall gain.

The primary objective of this study is to identify and assess materials capable of protecting the antenna from the lunar environment without compromising its performance. In the pursuit of suitable materials, a careful selection was made for the testing of coating materials applied to the sleeve dipole antenna. The chosen materials include white thermal control coatings (White TCC), acrylic paint, a polyimide film, aluminum foil, stainless steel, and zirconium alloy. White TCC is a material used for antennas due to its ability to withstand exposure to solar flux, radiations, extreme temperatures, and vacuum [33].

The second material used is acrylic paint, valued for its dual functionality due to its versatility within a moderate temperature range and low solar absorptance. Furthermore, it proves effective when applied to structures to control radiative heat transfer and is used on multilayer insulation blankets for sealing edges and addressing tears in the outer layers. A polyimide film with permittivity = 3.5, thermal conductivity K = 0.2 W/K/m, and specific heat = 1000 J/K/kg is incorporated into our research due to its capacity to withstand high-temperature environments, making it a suitable material for our study [34]. Due to its distinctive characteristics, aluminum foil was selected as the coating material for the sleeve dipole antenna. With the ability to reflect approximately 96% of radiant heat, it proves to be well-suited for thermal insulation, making it a convenient choice for coating purposes [35]. Moreover, the selection of stainless steel coatings is grounded in the reliability and efficiency of antenna coating technology. This choice is motivated by its capacity to prolong the lifespan of the component, providing robust protection against environmental threats [36]. The final selection for coating the sleeve dipole antenna is the zirconium alloy, chosen for its stability at excessive temperatures [37].

The aforementioned materials were chosen to safeguard the sleeve dipole antenna when deployed on the lunar surface, particularly against lunar environmental factors, including weather conditions.

### 4.2. All-Metal Patch Antenna Design

The patch antenna was chosen for its distinct and advantageous characteristics. Patch antennas provide several advantages, such as their low-profile design, simplified fabrication process, versatility to conform to both planar and non-planar surfaces, and seamless integration with integrated circuit technology [38,39,40]. The patch antenna installed on the Rashid rover enables the transfer of data from the Rashid rover to a base station located on Earth. A circularly polarized all-metal patch antenna was used for this purpose. As previously noted, in order to assess the effects of the lunar environment and rover structure, a patch antenna designed for facilitating successful communication between the rover and the lander was configured at the same frequency and polarization as the sleeve dipole antenna, ensuring consistency for comparative analysis. This design was executed using CST Microwave Studio, as illustrated in Figure 3. The circular patch antenna comprises entirely metallic components, with both the ground plane and the circular patch constructed from metal and connected through the center conductor of the connector. The simulation setup assumes ideal conditions where the patch is maintained in the correct position and orientation by this connector pin. Figure 3 displays various perspectives of the patch antenna. The circular metallic patch measures 65 mm in diameter and 0.5 mm in thickness. The ground plane has dimensions of 65 mm × 65 mm (length × width) and a thickness of 0.5 mm. A 0.6072 mm gap separates the circular patch from the ground.

### 4.3. Antennas Integrated into Rover Structure

During the second phase of the simulation, the antennas were incorporated into the rover structure, and their individual performances were assessed. This study holds significant importance in comprehending how different materials within the rover influence each antenna’s performance. Initially, a sleeve dipole antenna was installed on the rover, as illustrated in Figure 4, followed by the integration of a patch antenna atop the camera box. For the actual mission, tests were conducted to ensure that the camera box can serve as a ground plane for the antenna without affecting its performance. These tests confirmed that the camera box provides adequate support and does not interfere with the antenna’s functionality. For the simulation, the antennas were integrated separately, and their effects were studied individually, taking into account the operational principle that only one antenna will be activated at any given time during a real mission.

## 5. Results and Discussion

This section explains the significant observations derived from our simulation procedure. The results obtained effectively bridge gaps in theoretical understandings regarding antenna behavior in the lunar environment.

### 5.1. Performance of Standalone Antennas: Sleeve Antenna

In the examination of the performance of two antennas, namely the sleeve dipole antenna with a length of 55.7 mm and the sleeve dipole antenna covered by a tube with a length of 159.5 mm, it was observed that the reflection coefficient results were identical for both configurations [24]. Both antennas share an operational frequency of 2.42 GHz and exhibit an impedance bandwidth spanning 585 MHz, ranging from 2.18 GHz to 2.765 GHz, Figure 5a. At 2.42 GHz, both antennas manifest an omnidirectional radiation pattern, with the unmodified antenna achieving a peak gain of 2.27 dBi, and the antenna with a tube registering a peak gain of 2.23 dBi. It is noticed from the simulation results that the use of the tube in increasing the antenna’s length does not affect its performance. Theoretically, dipole antennas have nulls at their axes and the maximum occurs orthogonally [38]. In the case of the designed sleeve dipole antenna, the same behavior can be observed, as shown in Figure 5b.

The reflection coefficient comparison of the sleeve dipole antenna with different coatings is represented in Figure 6a. The primary coating of acrylic paint and the secondary coating of polyimide film demonstrate an ideal S_11_ matching for both the coated and uncoated sleeve antennas. Our previous study [25] demonstrated that the S_11_ characteristics of the sleeve dipole antenna remained relatively stable despite varying secondary coatings, such as aluminum foil, stainless steel, and zirconium alloy, applied after an initial layer of white thermal control coating (White TCC). Notably, negligible alterations were observed with both white TCC and zirconium alloy coatings, aligning with the expectations for standard protective coatings. However, it was also noticed that aluminum foil and stainless steel induce substantial performance changes, deviating from efficiency expectations. It was noticed that the radiation pattern of the sleeve dipole antenna with diverse coating materials, including acrylic paint and polyimide film, resulted in a peak gain of 2.23 dBi, as shown in Figure 6b. Conversely, employing stainless steel and aluminum foil as coatings on the sleeve dipole antenna led to a decline in gain to 0.341 dBi, signifying a decrease in antenna efficiency, as shown in [25].

### 5.2. Performance of Standalone Antennas: Patch Antenna

The patch antenna operates at a frequency of 2.39 GHz, boasting an impedance bandwidth spanning 80 MHz, covering the range from 2.35 GHz to 2.43 GHz, as illustrated in Figure 7. The radiation pattern of the patch antenna in the XZ-plane and YZ-plane is shown in Figure 8a,b. As depicted, the patch antenna exhibits a directive pattern with a maximum gain of 8 dBi.

### 5.3. Antennas’ Performance on Atlas Regolith

An examination was conducted to evaluate the effect of coated antennas under the influence of lunar regolith, mimicking characteristics of those observed in the Atlas crater. Throughout subsequent simulations, an antenna coated with a combination of acrylic paint and a polyimide film remained consistently used. The regolith in the Atlas crater has unique characteristics [30]. In the complete absence of water, the DC electrical conductivity spans from 10^−14^ S/m for lunar regolith to 10^−9^ S/m for lunar rocks at a temperature of 300 K in the absence of light [41]. In all simulations, the size of the lunar regolith was set as 1600 mm × 1600 mm × 1700 mm (Width × length × height). A previous study [24] demonstrates that placing the antenna in close proximity to the lunar regolith significantly impacts its performance. It is generally advised to position the antenna as far away from the regolith as possible. However, practical mission constraints often impose limitations. In this study, the antenna was positioned at the distance specified by the mission requirements.

Figure 9 illustrates a comparison of the reflection coefficient of the coated sleeve dipole antenna when placed on top of the lunar regolith versus the original standalone antenna. It is evident that placing the antenna over the regolith reduces the original bandwidth from 585 MHz to 420 MHz while maintaining similar performance across both conductivity scenarios. Figure 10a,b show the radiation pattern of the coated antenna while placed over the regolith at both conductivities. When considering a conductivity of 10^−9^ S/m, the gain amounts to 4.62 dBi, whereas with a conductivity of 10^−14^, the gain increases to 4.76 dBi. Also, the pattern starts to be distorted with the introduction of ripples, while the conductivity changes from 10^−9^ S/m to –10^−14^ S/m. These distortions are due to the ground reflections from the lossy regolith. Similar findings were observed in [23] where the authors used the geometrical theory of diffraction to predict the behavior of antennas in the presence of lunar regolith.

Testing the impact of Atlas regolith characteristics on the patch antenna is also crucial. In this phase, Atlas regolith with the same dimensional as well as material properties was incorporated into the simulation. Two distances of the antenna above the lunar regolith were simulated for testing: one actual distance (718 mm) at which the patch antenna is placed in the actual mission and another random distance (98 mm) that is very close to the regolith.

Figure 11 illustrates the reflection coefficients of the patch antenna corresponding to two distances. In contrast to the sleeve dipole antenna, where the bandwidth was notably affected, the patch antenna showed a more consistent performance while placed on top of the lunar regolith. Figure 12a,b illustrate the XZ-plane as well as the YZ-plane pattern corresponding to two different distances from the regolith. The lunar regolith is also shown in the picture for a better understanding of the scenario. The pattern in the YZ-plane obtains a more significant gain change at this time with a maximum gain of 8.26 dB. Notably, at a height of 718 mm, a portion of the radiation power is observed to be unabsorbed by the lunar regolith. However, as shown here, ripples started to form at this stage. Conversely, at a height of 98 mm, being the lowest antenna height, the entire power of the radiation pattern is absorbed. This is expected due to the fact that a good conductive property of the ground can reflect RF signals whereas a poor conductivity can result in significant attenuation and signal loss [23]. Hence as proved in the previous study, it is better to place the antennas away from the lunar regolith.

### 5.4. Antennas’ Performance on Rover Structure

During this phase, the dipole antenna was incorporated into the rover structure and subjected to simulation. Specifically, the antenna was affixed to an extension structure positioned 510 mm above the lunar surface. This extension was implemented to elevate the antenna as high as possible, thereby enhancing the likelihood of maintaining Line of Sight (LOS) communication. The rover’s mast, located at a distance of 290 mm from the dipole antenna, was a key consideration due to its role as a significant obstacle between the rover and the lander. It was imperative to assess the impact of the rover’s mast on antenna performance. Additionally, the influence of the top metallic panel on signal propagation was evaluated during performance testing. Figure 13a,b illustrate the S-parameters and radiation characteristics of the system. As depicted in the figure, the operational frequency of the system closely matched that of a standalone antenna, at 2.42 GHz. Nevertheless, the 10 dB bandwidth is varied, ranging from 2.256 GHz to 2.672 GHz, resulting in a bandwidth of 416 MHz. The mission team has validated, through various parameters, that the communication data rate will be low in the real mission context. This indirectly explains the fact that the presence of a rover mast or regolith can reduce the bandwidth which in turn reduces the data rate. The radiation pattern obtained is illustrated in Figure 13b, demonstrating a modification in the omnidirectional radiation of the antenna, with the primary lobe directed towards the rover mast.

The same process was also performed for the patch antenna by integrating the patch on the rover structure, atop the camera box. As shown in Figure 14a, the reflection coefficient and bandwidth of the patch antenna are less affected by the rover materials which emphasizes the robustness of patch antennas compared to the dipole antennas. However, as shown in Figure 14b, the pattern at the YZ-plane is modified and achieves a similar shape to that of the XZ-plane with a maximum gain of 8.9 dBi.

#### Rotations of the Patch Antenna on Atlas Regolith

To ensure effective communication between the Rashid rover in the Atlas crater and ground stations on Earth, a pointing mechanism for the antenna is required. Through simulations, we accurately studied the angles of elevation (ALT) and azimuth (AZ) of both the Sun and Earth at the Atlas prime landing site throughout the surface mission duration, spanning from 24 April to 9 May 2023. The analysis revealed the dynamic movement of Earth relative to the lunar surface during the operational phase. To capitalize on the available communication windows and to maximize the transfer of energy between the rover’s patch antenna and the designated Earth stations, a robust pointing mechanism becomes necessary. Figure 15 illustrates the fluctuating positioning of Earth, reinforcing the necessity for a flexible and responsive antenna pointing system to maintain uninterrupted communication links during the lunar mission. Consequently, our analysis indicated that the elevation angle would be particularly low at the commencement and conclusion of the surface operation. This observation underscores the critical importance of studying the effect of lunar regolith on antenna performance.

In this context, this section investigates the concept of antenna rotation, aimed at orienting the antenna towards Earth. Three rotational scenarios were simulated: a 90° clockwise rotation (antenna rotated towards the right), a 90° counterclockwise rotation (antenna rotated towards the left), and a 180° rotation where the antenna’s patch faces the lunar regolith. Rotating the antenna introduces the possibility of the lunar regolith’s conductivity influencing the antenna’s operating frequency and radiation pattern in diverse manners. The simulation results will present these variations, along with any similarities that may emerge. In the subsequent simulations, the regolith conductivity remains at 10^−9^ S/m, while the antenna height is set at 335 mm above the regolith. Across all rotation scenarios, the operational frequency remains constant at 2.39 GHz (Figure 16a). However, both the radiation pattern and the gain are influenced by the rotation. The 90° clockwise and counterclockwise rotations yield a gain of 7.3 dBi, with the radiation patterns for both scenarios appearing nearly identical. Conversely, the 180° rotation results in a gain of 12.4 dBi, with the majority of the radiation pattern being absorbed by the lunar regolith (Figure 16b,c).

### 5.5. Antennas’ Performance on Rover Structure Placed on Top of Atlas Crater

In this phase, the Rashid rover structure was incorporated into the design. The objective was to examine the performance of the sleeve and patch antenna in the rover structure and lunar regolith, characterized by two different conductivities: 10^−14^ S/m and 10^−9^ S/m [41]. First, the sleeve dipole antenna was mounted atop the Rashid rover structure, situated above the lunar regolith. Analyses indicate that the antenna sustains a consistent operational frequency of 2.42 GHz across both conductivity scenarios of 10^−9^ S/m and 10^−14^ S/m, as depicted in Figure 17a. Both the operating frequency and bandwidth remain nearly identical for both conductivities, which is the same as the obtained bandwidth while integrating the antenna on the rover. Nevertheless, variations in the electrical characteristics of the regolith result in marginal alterations in both gain and radiation patterns. Specifically, a conductivity of 10^−9^ S/m yields a gain of 4.56 dBi at the YZ-plane, whereas a conductivity of 10^−14^ S/m results in a gain of 4.44 dBi. Corresponding gains at the XZ-planes are 3.37 dBi and 3.52 dBi, respectively. The obtained radiation patterns are illustrated in Figure 17b,c.

Later, the patch antenna was mounted atop the Rashid rover structure, situated above the lunar regolith. In this simulation, the sleeve antenna was removed to see the effect of the patch antenna only. Results prove that the antenna sustains a consistent operational frequency as well as bandwidth across both conductivity scenarios of 10^−9^ S/m and 10^−14^ S/m, as depicted in Figure 18a. It is evident that a patch antenna is less vulnerable to environmental conditions than a sleeve antenna. However, as previously mentioned, variations in the electrical characteristics of the regolith result in marginal alterations in both gain and radiation patterns. Specifically, a conductivity of 10^−9^ S/m yields a gain of 9.9 dBi at the YZ-plane, whereas a conductivity of 10^−14^ S/m results in a gain of 9.8 dBi in the XZ-plane. The gains at the YZ-planes are 9.5 dBi and 9.47 dBi, respectively. The corresponding radiation patterns are illustrated in Figure 18b,c. Table 1 summarizes the key parameters of lunar antennas reported in the literature.

## 6. Discussion

Lunar missions have garnered significant attention in recent years. Many countries, including the United States, China, and India, have solidified their positions in lunar exploration, while others, such as the United Arab Emirates, have declared their intentions to embark on lunar missions soon. The Moon has become a technological hub for research and development. Given the significance of these missions, understanding the lunar environment prior to their initiation is crucial. Antennas are pivotal components of the communication system, acting as sensors for the entire network. Therefore, the stability of these antennas, both electrical and mechanical, is essential. Conducting electromagnetic simulations that replicate the constraints imposed by the lunar environment is a fundamental step in this regard. Although antennas suitable for space applications are widely documented in the literature, antennas specifically designed for lunar applications are limited or unavailable to the research community. The main reason is that knowledge about specific design constraints is often known only to mission agencies. Studying stand-alone antennas that are capable of producing the necessary gain, bandwidth, and polarization is not sufficient to meet mission requirements. Mission-specific antennas may require additional coatings or special materials to withstand the extreme temperature variations on the lunar surface. Unlike terrestrial ground, the lunar ground is lossy, and its material properties vary from place to place. It is crucial to include the lunar regolith in electromagnetic simulations with the exact properties of the landing site to understand how it affects antenna behavior in terms of operating frequency, bandwidth, gain, and radiation pattern. Additionally, antennas must be integrated into the rover structure, including the mast configuration and material composition. Since a rover mast may be in close proximity to the antenna, it is important to understand how it affects the antenna’s performance. Understanding the complete setup, including the rover structure and lunar regolith, is also crucial. Insights gained from these simulations will help future missions. Our simulation study, conducted with an exact replica of the lunar Rashid rover structure and assigned material properties for the regolith at the landing site, showed that regolith and the rover mast can affect the gain and radiation pattern of the antenna. We observed that placing the antenna close to the regolith creates ripples in the pattern, resulting in signal attenuation and losses. Integrating the sleeve dipole antenna into the rover reduced the bandwidth from 585 MHz to 416 MHz. However, the patch antenna was less vulnerable to the rover structure. A field test between the rover and the lander, involving rover mast, was conducted with the rover positioned at various angles relative to the lander. The purpose was to observe the impact on the data transfer rate of the communication link. The results showed that the data rate varied with different angles, sometimes increasing and sometimes decreasing. This variation supports the conclusion that the mast affects antenna performance, confirming the results of prior simulations. Given the difficulty of making adjustments after mission launch, it is essential to consider these constraints to ensure reliable communication.

## 7. Conclusions

In this paper, a systematic approach to explore how lunar regolith and rover structure, including mast configuration and material composition, impact antenna parameters has been presented. In practical missions, antennas are often integrated into rovers made of diverse materials. Understanding antenna performance when positioned on lunar regolith beneath a rover is critical. Our findings indicate that both lunar regolith and rover structure can modify antenna radiation properties. Therefore, careful consideration should be given to antenna placement on rover structures. Additionally, we observed that antenna characteristics can be influenced by lunar regolith, with less impact observed when antennas are positioned away from lossy regolith. Furthermore, our study concludes that patch antennas demonstrate less vulnerability to environmental conditions than the sleeve antenna. The general conclusions from this study can be applied to mission requirements such as the distance between the antenna and the regolith, the influence of different coating materials on antenna performance, and the impact of lossy regolith. In general, for the Rashid mission, this study enables the design of antennas that maximize the communication range, data transmission rates, and power efficiency, while mitigating interference and signal attenuation.

## Figures and Tables

**Figure 1 sensors-24-05361-f001:**
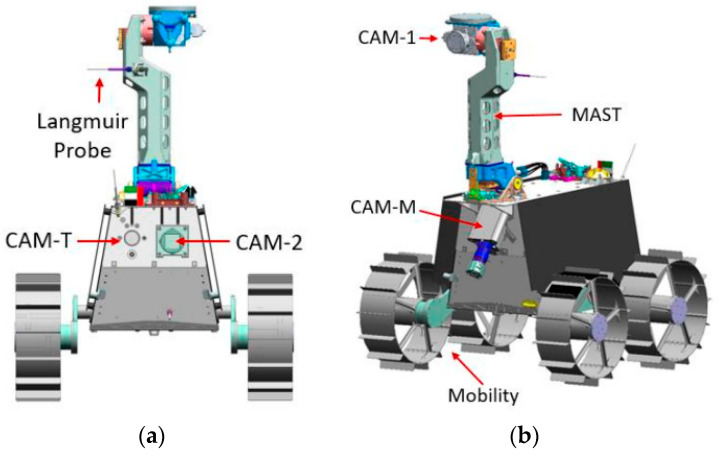
(**a**) Rashid rover back view, (**b**) Rashid rover three-dimensional view.

**Figure 2 sensors-24-05361-f002:**
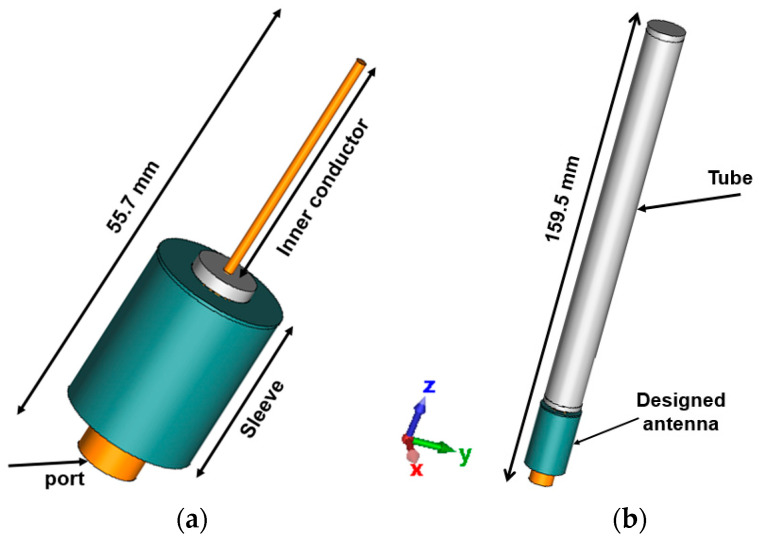
Designed sleeve dipole antenna, (**a**) sleeve dipole antenna, (**b**) sleeve dipole antenna with tube.

**Figure 3 sensors-24-05361-f003:**
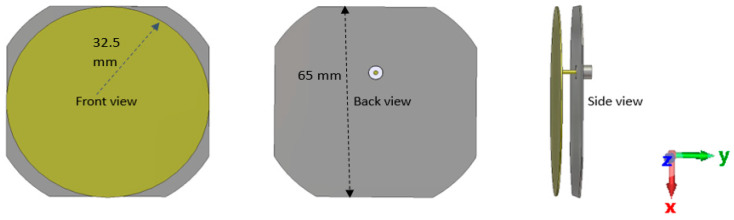
Designed patch antenna.

**Figure 4 sensors-24-05361-f004:**
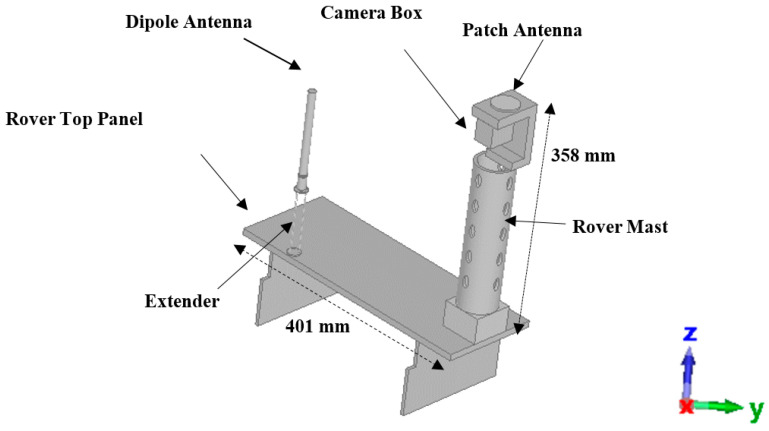
Simulated model of Rashid rover with sleeve dipole antenna and patch antenna.

**Figure 5 sensors-24-05361-f005:**
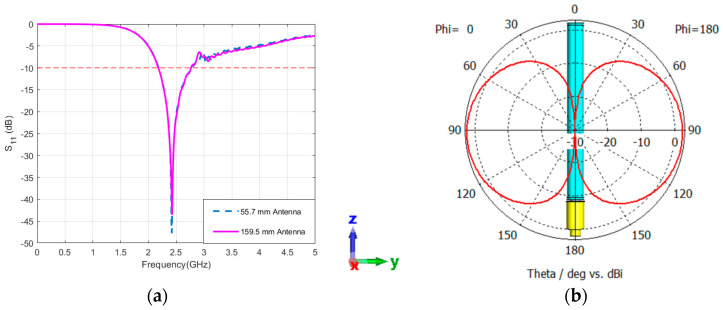
Simulated characteristics of the sleeve dipole antenna. (**a**) Reflection coefficient for the 55.7 mm antenna and the 159.5 mm antenna with a tube, (**b**) corresponding radiation pattern at the YZ-plane. The coordinates represent the orientation of the antenna.

**Figure 6 sensors-24-05361-f006:**
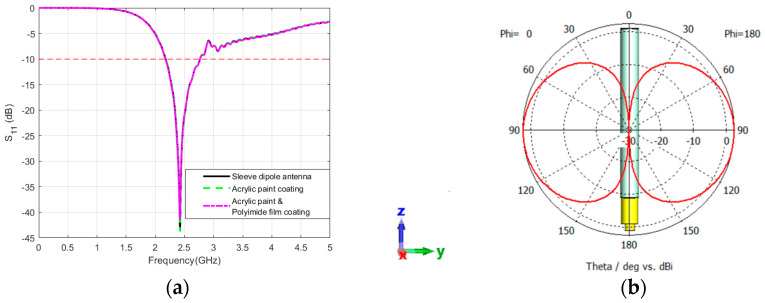
(**a**) Reflection coefficient (S_11_) for the sleeve dipole antenna with acrylic paint and polyimide film coating. (**b**) Radiation pattern (YZ-plane) for the sleeve dipole antenna with different material coatings. The coordinates represent the orientation of the antenna.

**Figure 7 sensors-24-05361-f007:**
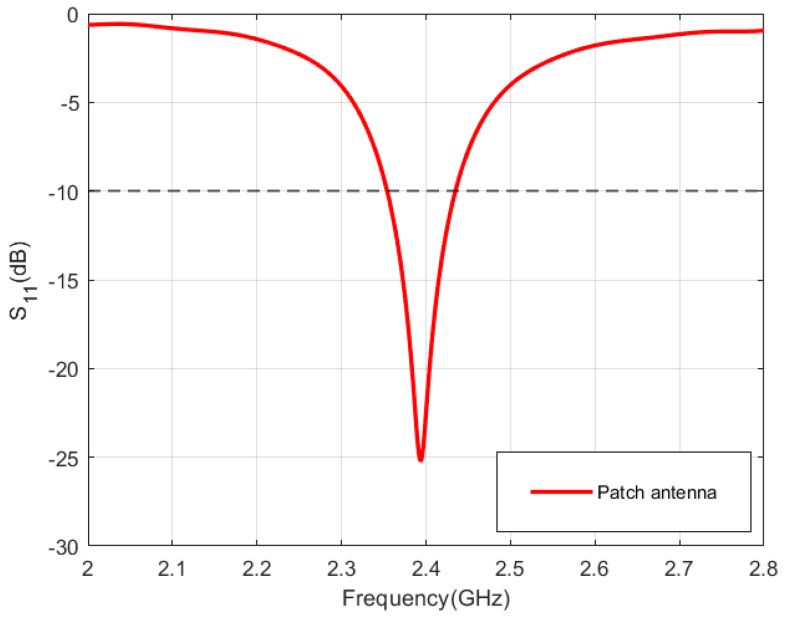
Reflection coefficient (S_11_) of the patch antenna.

**Figure 8 sensors-24-05361-f008:**
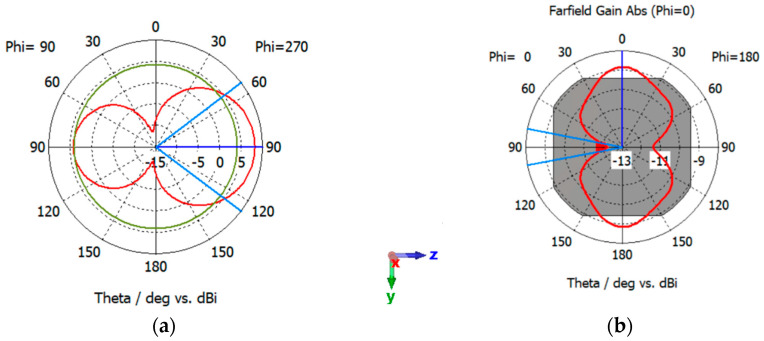
Radiation pattern of the patch antenna. (**a**) XZ-plane, (**b**) YZ-plane. The coordinates represent the orientation of the antenna.

**Figure 9 sensors-24-05361-f009:**
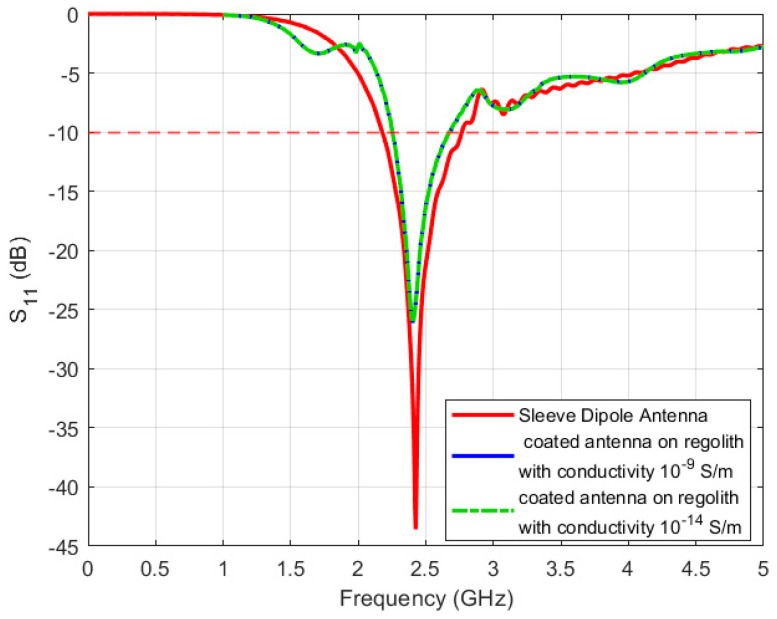
Reflection coefficient of the sleeve dipole antenna for the conductivities 1 × 10^−9^ and 1 × 10^−14^.

**Figure 10 sensors-24-05361-f010:**
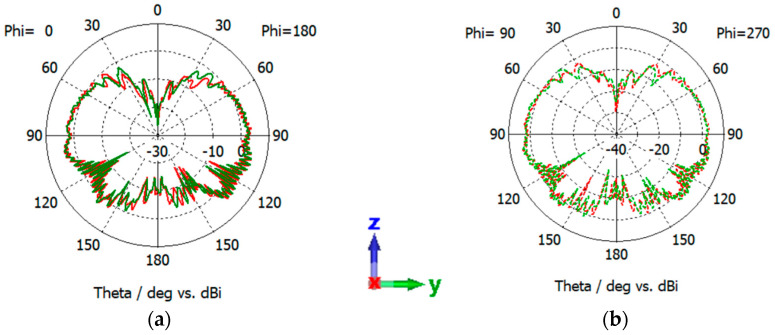
Radiation pattern of the sleeve dipole antenna for the conductivities 1 × 10^−9^ (red color) and 1 × 10^−14^ (green color). (**a**) YZ-plane, (**b**) XZ-plane. The coordinates represent the orientation of the antenna.

**Figure 11 sensors-24-05361-f011:**
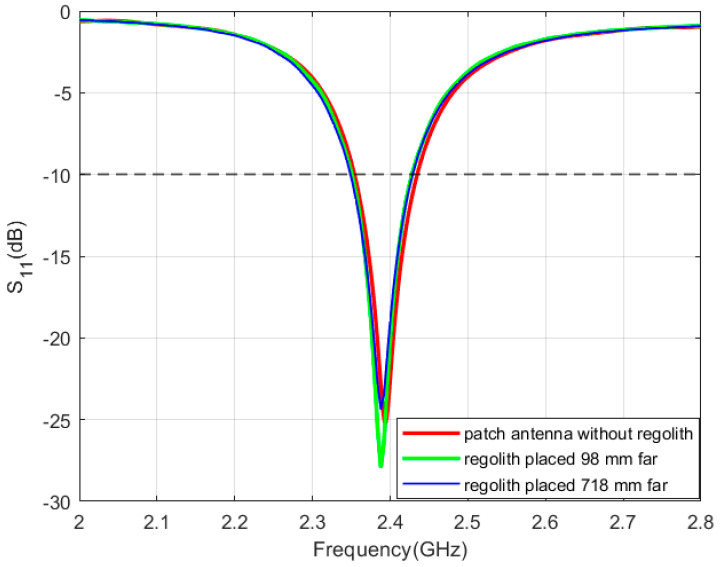
Reflection coefficient of the patch antenna at different heights from lunar regolith.

**Figure 12 sensors-24-05361-f012:**
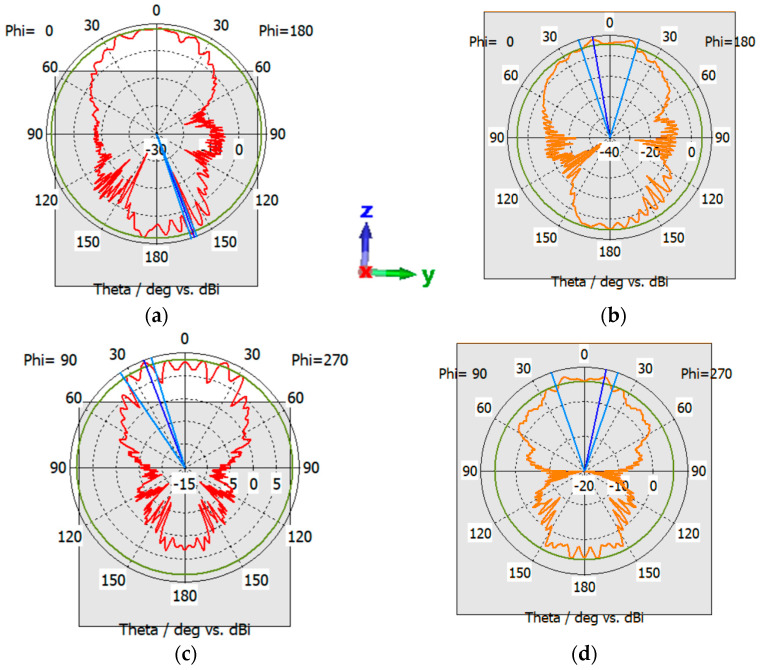
Radiation pattern of the patch antenna at two different heights, 718 mm (red curves) and 98 mm (orange curves), from lunar regolith at the XZ- as well as YZ-planes. (**a**) Pattern at the XZ-plane when the antenna is at a height of 718 mm. (**b**) Pattern at the XZ-plane when the antenna is at a height of 98 mm. (**c**) Pattern at the YZ-plane when the antenna is at height of 718 mm and (**d**) pattern at the YZ-plane when the antenna is at a height of 98 mm. The coordinates represent the orientation of the antenna.

**Figure 13 sensors-24-05361-f013:**
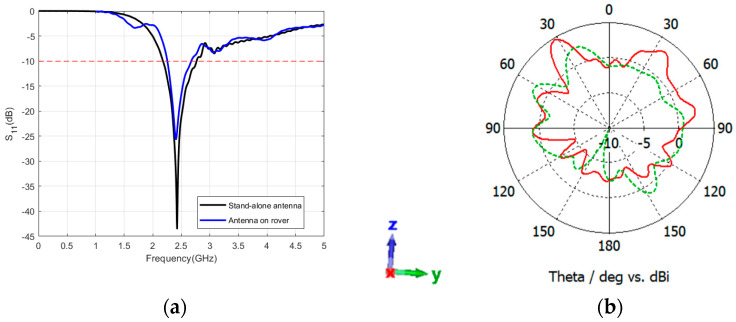
Reflection coefficient (**a**) and radiation pattern (**b**) of the sleeve dipole antenna while integrated with the rover structure. The red color shows the pattern at the XZ-plane and the green color shows the pattern at the YZ-plane. The coordinates represent the orientation of the antenna.

**Figure 14 sensors-24-05361-f014:**
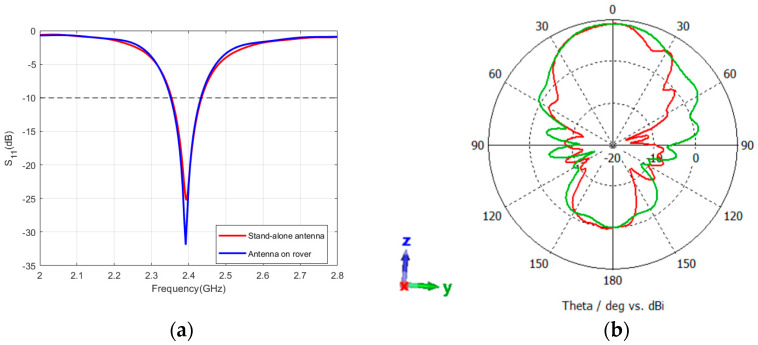
Reflection coefficient (**a**) and radiation pattern (**b**) of the patch antenna while integrated with the rover structure; the red color shows the pattern at the XZ-plane and the green color shows the pattern at the YZ-plane. The coordinates represent the orientation of the antenna.

**Figure 15 sensors-24-05361-f015:**
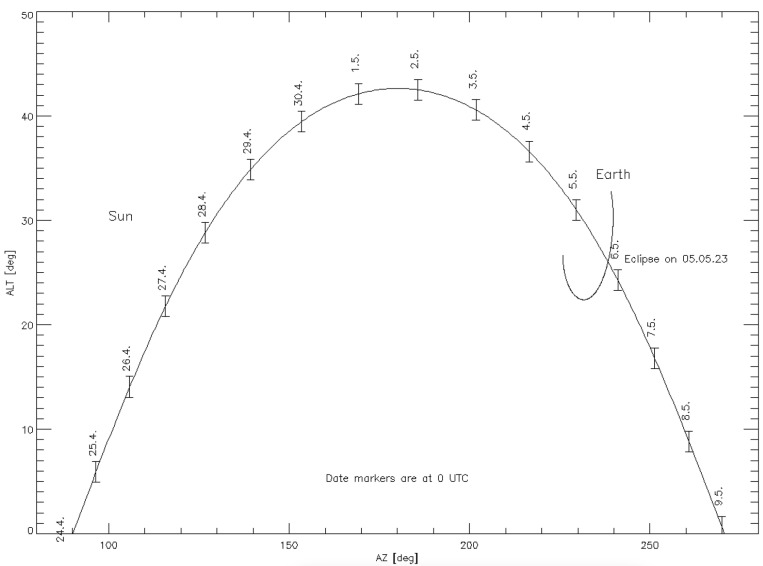
Lunar day in Atlas crater, 24 April to 9 May 2023. ALT is the elevation and AZ is the azimuth at the prime landing site.

**Figure 16 sensors-24-05361-f016:**
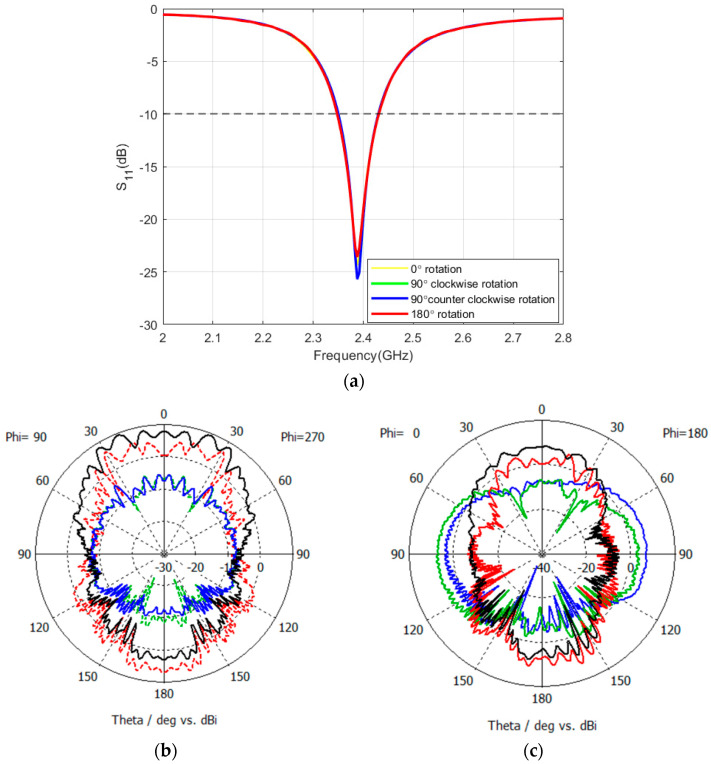
Reflection coefficient and radiation pattern of the patch antenna for a 90° clockwise (blue color), 90° counterclockwise (green color), and 180° antenna patch facing the regolith (red color). (**a**) reflection coefficient (**b**) pattern at the XZ-plane, (**c**) pattern at the YZ-plane.

**Figure 17 sensors-24-05361-f017:**
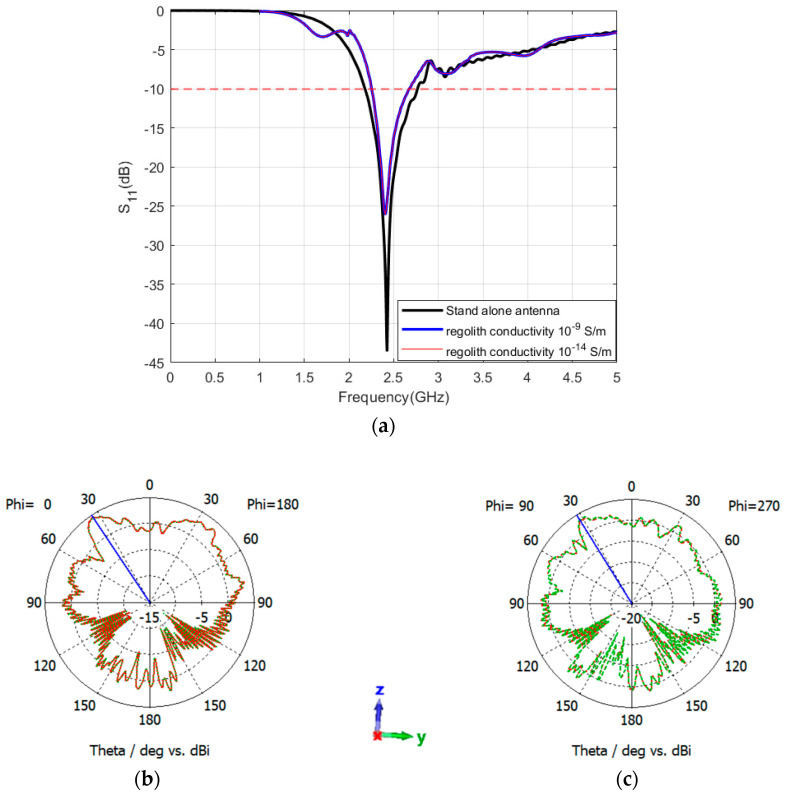
Reflection coefficient and radiation pattern of the sleeve dipole antenna integrated with the rover placed on top of Atlas crater for conductivities of 1 × 10^−9^ (red color) and 1 × 10^−14^ (green color). (**a**) reflection coefficient (**b**) XZ-plane (**c**) YZ-plane. The coordinates represent the orientation of the antenna.

**Figure 18 sensors-24-05361-f018:**
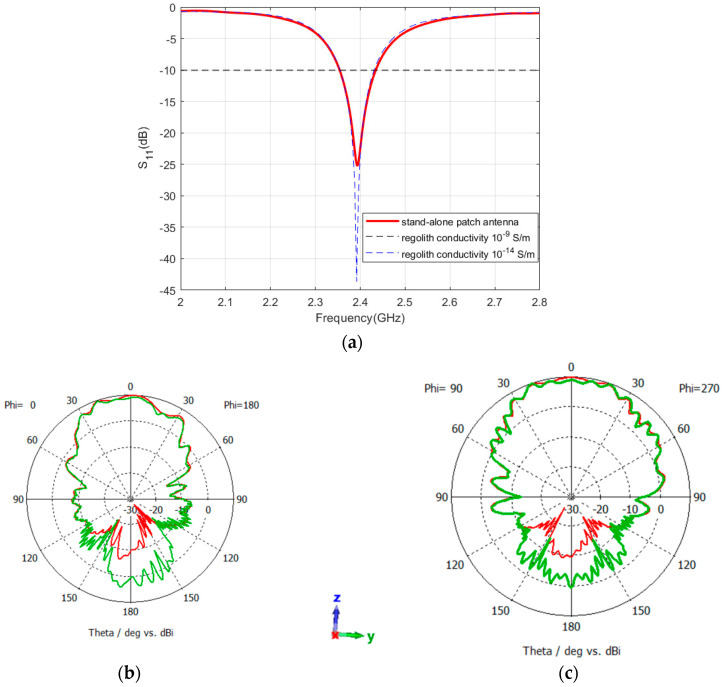
Reflection coefficient and radiation pattern of the patch antenna integrated with the rover placed on top of Atlas crater for conductivities of 1 × 10^−9^ (red color) and 1 × 10^−14^ (green color). (**a**) reflection coefficient, (**b**) XZ-plane (**c**) YZ-plane. The coordinates represent the orientation of the antenna.

**Table 1 sensors-24-05361-t001:** Comparison of lunar antennas in terms of key constraints.

Ref.	Antenna Name	Objective	Design Type	Key Lunar Constraints Considered in This Study	Simulation Tool	Application
[16]	Microstrip monopole antenna	Research purpose	Stand-alone antenna	Miniature antenna size	Zeland’s IE3D electromagnetic simulator	Lunar surface communications
[16]	Fractal antenna	Research purpose	Stand-alone antenna	Miniature antenna size	Zeland’s IE3D electromagnetic simulator	Lunar surface communications
[17]	Vivaldi antenna	Real mission	Antennas on lander	Coating, lander	n/a	Lunar regolith penetrating radar
[18]	Channel 1 antenna	Real mission	Antenna on rover	Rover, coating	FEKO	Lunar penetrating radar
[18]	Channel 2 antenna	Real mission	Antenna on rover	Rover, coating	FEKO	Lunar penetrating radar
[19]	Inflatable impulse radiating antenna	Research purpose	Stand-alone antenna	Suitable thin-film materials	FEKO	Radar
[20]	Backfire antenna	Research purpose	Stand-alone antenna	All-metal antenna	Ansys HFSS	Communication and navigation
[22]	Dipole antenna	Real mission	Antenna arrays on radio observatory	Coating, terrestrial ground, extreme temperature, radio observatory system	CST	Near-side low radio frequency imaging
[23]	Dipole antenna	Research purpose	Stand-alone antenna	Regolith, crater terrain	Geometrical Theory of Diffraction	Lunar wireless communication and sensor systems
[42]	Small-sized active dipoles	Research purpose	Stand-alone antenna (active antenna)	Solar U + III bursts	n/a	Lunar radio telescopes
[This work]	Sleeve dipole and all-metal patch antennas	Real mission	Antennas on rover placed over regolith	Rover, coating, and regolith	CST	Lunar surface communication

## Data Availability

Data are contained within the article.

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
