# Peer review of "Investigating the Impact of Lunar Rover Structure and Lunar Surface Characteristics on Antenna Performance"

_sensors, 2024, doi:10.3390/s24165361_

Round 1

Reviewer 1 Report (Previous Reviewer 2)

Comments and Suggestions for Authors

Comments:

The main concern about the manuscript is the lack of validation for the simulation results. Only one simulation tool was used, and there are no measured results or alternative numerical techniques to demonstrate the validity of the results. For example, if a simulation parameter was set incorrectly, how would this be identified?

Comments on the Quality of English Language

The paper requires minor English review.

Author Response

 We made every effort to address all the questions raised. Attaching the response file. As this study is part of the ongoing lunar mission, we are unable to publish experimental details at this stage. However, our conclusions are indirectly validated by existing literature and the studies conducted by the mission team. We hope this article will serve as a systematic guide for antenna designers interested in developing antennas for lunar applications. 

Reviewer 2 Report (New Reviewer)

Comments and Suggestions for Authors

In this work, using EM simulations results, the authors have examined possible influence factors on two specific antennas performance, planned to be used on a specific mission of a Lunar Rover. They investigated factors as lunar regolith, rover structure and materials used to cover the antennas. 

The novelty of the paper is rather weak. Findings are almost expected. The main contribution of the work is to gather information and references about materials used and environmental conditions related to such a lunar mission.

It is needed to add some measurement results to evaluate the simulation findings.

It could be interested to give more data of the link budget calculations both for the primary and the secondary antenna.

In Figure 12: denote in the figure legend the colors of the patterns: (green, red) for the corresponding heights

Author Response

 We made every effort to address all the questions raised. Attaching the response file. As this study is part of the ongoing lunar mission, we are unable to publish experimental details at this stage. However, our conclusions are indirectly validated by existing literature and the studies conducted by the mission team. We hope this article will serve as a systematic guide for antenna designers interested in developing antennas for lunar applications. 

Reviewer 3 Report (New Reviewer)

Comments and Suggestions for Authors

Comments

This article explores the influence of lunar regolith and rover structure, such as mast design and material composition, on antenna parameters. Lunar regolith is a layer of unconsolidated rock material that covers bedrock. The rover is expected to operate in the Moon region known as Atlas Crater. The regolith is thinnest within Atlas crater (~1.6 m, on average).

The regolith layer is heterogeneous, so as the rover moves through the crater, the thickness of the regolith underneath will change; it is possible that in some places it will be 0.6 m, and in others – 2.6 m. Under the regolith, there is solid bedrock, which physical parameters differ sharply from the regolith. The rover's communications equipment operates at frequencies close to 2 GHz, i.e. the wavelength in free space is about 15 cm. Thus, the distance from the regolith surface to its boundary with granite can be only 4 wavelengths. Recently published works ( DOI: 10.1109/ACCESS.2023.3294694) have shown that the parameters of antennas over two-layer lunar soil differ markedly from the antenna parameters over homogeneous regolith. Therefore it is necessary to take into consideration the effect of the interface regolith/bedrock on the antenna parameters.

 Based on the above, I would like to make the following recommendations to the authors. 

1. The article must indicate exactly what a lunar soil model was used to calculate the antenna parameters, Homogeneous half-space with the exact Sommerfeld solution, Homogeneous half-space with reflection coefficient approximation, or another.

2. It is necessary to justify why the authors did not use a two-layer model of lunar soil, which could more accurately predict the parameters of the rover antennas.

Author Response

 We made every effort to address all the questions raised. Kindly find the attached  response file. 

Round 2

Reviewer 1 Report (Previous Reviewer 2)

Comments and Suggestions for Authors

The authors have applied the suggestions proposed by this reviewer.

This manuscript is a resubmission of an earlier submission. The following is a list of the peer review reports and author responses from that submission.

Round 1

Reviewer 1 Report

Comments and Suggestions for Authors

Author Response

We appreciate your time and effort in reviewing the article. We made every effort to thoroughly address each question. Below are the comments (black) and their corresponding responses (red). Text in blue indicates additions made to the article.

Reviewer 2 Report

Comments and Suggestions for Authors

COMMENTS:

The submitted manuscript presents an analysis of the influence of the rover structure and lunar regolith on the performance of a sleeve dipole and a microstrip patch antenna in the context of lunar missions. This is a topic of significant interest to the antennas and propagation community. However, the manuscript requires a thorough review. Some of the points that need attention are detailed below.

1) The Introduction mainly refers to papers [16] and [17], written by some of the manuscript’s contributors, to examine the impact of lunar rover structure and surface characteristics on antenna performance. However, the literature review should be broadened to include a wider range of previous studies. By incorporating a more comprehensive review, the manuscript can better emphasize its original contributions and the significance of its results. The authors need to clarify the original contributions and the relevance of the results in the manuscript.

2) Line 127: A period seems missing after reference [19].

3) Line 144: Figure 1 is enough.

4) Line 174: It should be kHz.

5) Please check the correctness of the word “utilized” throughout the manuscript.

6) Since the patch antenna has only one patch, it should not be classified as a stacked configuration. Microstrip antennas with more than one patch, mounted with their centers vertically aligned, are usually classified as stacked patches.

7) Section 4.2: Have the authors considered a structure to support the patch? The connector’s center pin alone is probably insufficient to support the patch.

8) Section 4.2: It is stated that “A circularly polarized all-metal patch antenna was used for this purpose.” However, why have the authors analyzed a linearly polarized patch antenna operating at the same frequency and polarization as the sleeve dipole?

9) Section 5.1: What is the tube thickness of the tube used to cover the sleeve dipole? Additionally, what are the relative permittivity and loss tangent values for ABS used in the CST simulation?

10) Lines 309 and 310: Review the statement in these lines because Figure 6(a) does not show a cut view of the sleeve dipole.

11) Section 5.1: The relative permittivity, loss tangent, and thickness of the outer coatings should be described in the manuscript.

12) Lines 312 to 323: The conclusions in these lines are not supported by the graphs in Figure 6.

13) Lines 336 and 337: There is a formatting problem.

14) Section 5.2: Why did the authors choose a ground plane with dimensions close to those of the patch?

15) Figure 8: Please check the pattern in Figure 8(b), as its maximum gain is approximately –9 dBi. Is this really expected?

16) Lines 367 and 368: The authors report a gain increase of approximately 0.1 dB. This increase is significantly smaller than the gain uncertainty associated with many methods used to measure antenna gain, as well as the link margins typically considered in a link budget analysis. As such, the reported difference in gain may not represent a practical change in system performance.

17) The authors should justify their classification of “distortions” in the patterns of Figure 10. Are the ripples observed in the patterns the result of edge diffractions since the regolith area is limited?

18) What solver was used to analyze the antennas in CST? Was a hybrid simulation carried out for the analysis including the rover? Furthermore, how have the authors validated the results presented in the manuscript, considering that no experimental data are shown? Have they checked the results using another solver or simulation tool?

Comments on the Quality of English Language

The quality of the English language is good, and minor editing is required. Please check the use of the word "utilized".

Author Response

(The authors gave the same response as above.)

Round 2

Reviewer 1 Report

Comments and Suggestions for Authors

I think the current form is appropriate for application. No further comments.

Author Response

Thank you for your valuable time. 

Reviewer 2 Report

Comments and Suggestions for Authors

COMMENTS:

The authors have not addressed all the issues raised by this reviewer. The following items retain the numbers from the first review and describe the unanswered points.

7) The question posed in this item is relative to the patch and not the entire antenna. How is the patch of the microstrip antenna supported? Is it only by the connector’s center pin?

9) The tube thickness of 200 microns is not mentioned in the text. Have the authors checked whether such a thin ABS layer possesses adequate mechanical strength for the application, thereby justifying its electromagnetic analysis?

11) The electrical characteristics of the outer coatings were not added to the text.

15) The XYZ coordinate system should be included in the figures depicting the antenna geometries. Without this, interpreting the patterns shown in the document is difficult.

18) If the authors have data that validate the simulation results, they should present them in the manuscript. This would underscore the scientific soundness of the conclusions drawn in the text.

Additional points:

A) The range of 115 dB in the patterns in Figs. 5 and 6 is not practical.

B) The literature review should be broadened to include publications in journals and magazines, as many references in the current version were published in conferences and symposiums.

Comments on the Quality of English Language

The quality of the English language is good, and minor editing is required.

Author Response

Thank you for your valuable time. We tried our best to answer all the questions. Kindly find the attached file. 
